

# OLA1 is responsible for normal spindle assembly and SAC activation in mouse oocytes

Di Xie[1,2], Juan Zhang[2], JinLi Ding[1], Jing Yang[1] and Yan Zhang[3]

[1] Reproductive Medical Center, Renmin Hospital of Wuhan University, WuHan, HuBei, China
[2] Reproductive Medical Center, Central Theater General Hospital of PLA, WuHan, HuBei, China
[3] Department of Clinical Laboratory, Renmin Hospital of Wuhan University, WuHan, HuBei, China

## ABSTRACT

**Background**. OLA1 is a member of the GTPase protein family; unlike other members, it possess both GTPase and ATPase activities, and can bind and hydrolyze ATP more efficiently than GTP. OLA1 participates in cell proliferation, oxidative response, protein synthesis and tumorigenesis. However, whether OLA1 is also required for oocyte meiosis is still unknown.

**Methods**. In this study, the localization, expression, and functions of OLA1 in the mouse oocyte meiosis were examined. Immunofluorescent and confocal microscopy were used to explore the location pattern of OLA1 in the mouse oocyte. Moreover, nocodazole treatment was used to confirm the spindle-like location of OLA1 during mouse meiosis. Western blot was used to explore the expression pattern of OLA1 in the mouse oocyte. Microinjection of siRNA was used to explore the OLA1 functions in the mouse oocyte meiosis. In addition, chromosome spreading was used to investigate the spindle assembly checkpoint (SAC) activity.

**Results**. Immunofluorescent staining showed that OLA1 evenly distributed in the cytoplasm at germinal vesicle (GV) stage. After meiosis resumption (GVBD), OLA1 co-localized with spindles, which was further identified by nocodazole treatment experiments. Knockdown of OLA1 impaired the germinal vesicle breakdown progression and finally resulted in a lower polar body extrusion rate. Immunofluorescence analysis indicated that knockdown of OLA1 led to abnormal spindle assembly, which was evidenced by multipolar spindles in OLA1-RNAi-oocytes. After 6 h post-GVBD in culture, an increased proportion of oocyte which has precociously entered into anaphase/telophase I (A/TI) was observed in OLA1-knockdown oocytes, suggesting that loss of OLA1 resulted in the premature segregation of homologous chromosomes. In addition, the chromosome spread analysis suggested that OLA1 knockdown induced premature anaphase onset was due to the precocious inactivation of SAC. Taken together, we concluded that OLA1 plays important role in GVBD, spindle assembly and SAC activation maintenance in oocyte meiosis.

Corresponding authors
Jing Yang, dryangqing@hotmail.com
Yan Zhang,
peneyyan@mail.ustc.edu.cn

## INTRODUCTION

Mammalian gametes are yielded through an event named meiosis where two consecutive divisions are conducted to halve the chromosomes without an intervening replicative process (*Verlhac & Terret, 2016*). Spindle assembly and chromosome segregation are both vital courses to keep genomic stability during oocyte meiosis. Errors in these two processes can lead to the failure of meiosis or the generation of aneuploidy (*Ma et al., 2014*; *Jiao et al., 2017*; *Han et al., 2015*). Moreover, human pregnancy loss has strong correlation with fetal aneuploidy which can be induced by defective spindle structure and abnormal chromosome segregation (*Webster & Schuh, 2017*). To guarantee the normal progression of oocyte meiosis, spindle assembly and chromosome segregation must be correctly commanded.

After germinal vesicle breakdown (GVBD), as absent of centrosome at the mitosis, the meiotic spindle is emanated and nucleated from microtubule organizing centers (MTOCs) (*Schuh & Ellenberg, 2007*; *Clift & Schuh, 2015*; *Bennabi, Terret & Verlhac, 2016*). At this time, chromosomes and spindle microtubules has begun to interact with each other. Chromosomes continuously move to the equatorial plate as the cell cycle progresses, and the alignment of all chromosomes in this process does not coincide at the same time. Spindle assembly checkpoint (SAC) signaling controls this unevenly process to arrest oocyte at pre-metaphase I stage to avoid the premature chromosome segregation (*Vogt et al., 2008*; *Touati & Wassmann, 2016*; *Sanders & Jones, 2018*). Once the correct connection of chromosome with microtubule is established, all chromosomes are correctly aligned at the metaphase I plate (MI stage) then SAC proteins will dissociate from kinetochore to trigger the segregation of chromosomes (anaphase onset), which finally bring out the polar body extrusion and oocyte will arrest at metaphase II (MII).

OLA1 is a member of the GTPase protein family, unlike other members, it can bind and hydrolyze both ATP and GTP (*Koller-Eichhorn et al., 2007*). OLA1 can act as a negative regulator of the antioxidative response via nontranscriptional mechanisms (*Zhang et al., 2009*). Moreover, OLA1 involves in eukaryotic initiation factor 2 (eIF2)-mediated protein synthesis (*Chen et al., 2015*; *Ding et al., 2016*). During mitosis, OLA1 localizes to centrosomes in interphase and then to the spindle pole after nuclear envelop breakdown. OLA1 can directly interact with BRCA1 and γ-tubulin by bounding to the amino-terminal of these two proteins, and this interaction is very important for the centrosomal regulation (*Matsuzawa et al., 2014*). Knockdown of OLA1 results in the centrosome amplification of centrosome and the disorders in microtubule aster formation (*Matsuzawa et al., 2014*; *Yoshino et al., 2018*). It is noteworthy that BRCA1 has been shown to exert functions in oocyte meiosis, and knockdown of BRCA1 in mouse oocyte causes abnormal spindle assembly and the dysfunction of SAC (*Xiong et al., 2008*). As an interacting factor, whether OLA1 also participates in meiotic progression remains elusive. Thus, we attempted to explore the possible roles of OLA1 in oocyte meiosis.

Oocyte maturation is a complicated process, which can be affected by many factors including functional cellular proteins, and whether OLA1 is involved in meiosis is totally unknown. In this study, we have found that OLA1 participates in oocyte meiotic maturation,

especially in the progress of germinal vesicle breakdown, spindle assembly and SAC activation.

## MATERIALS AND METHODS

### Animals
Three-four weeks-old female KM mice were used in this experiment. Animal experiments were approved by Hubei Research Center of Laboratory Animal (Approval ID: SYXK (Hubei) 2014-0082SCXK). Animal care and handling were conformed to regulations of Animal Care and Use Committee of Central Theater General Hospital of PLA.

### Antibodies and chemicals
All chemicals and culture media were purchased from Sigma (St Louis, MO) unless those specifically mentioned. Rabbit polyclonal anti-OLA1 antibody (Cat# A4673) was purchased from ABclonal (Wuhan, China). Mouse monoclonal anti-α-tubulin-FITC antibody (Cat# F2168) was obtained from Sigma (St Louis, MO); Sheep polyclonal anti-BubR1 antibody (Cat# 28193) was obtained from Abcam (Cambridge, UK). FITC-conjugated donkey anti-sheep IgG (H + L) was purchased from Jackson ImmunoResearch Laboratory.

### Oocyte collection and culture
To harvest fully grown GV oocyte, 3–4 weeks-old female KM mice were firstly intraperitoneally injected with 5 IU pregnant mares serum gonadotropin, after 46–48 h, mice were sacrificed by cervical dislocation and ovaries were isolated. Enclosed cumulus oocytes were removed by repeatedly pipetting, and then oocytes were cultured in pre-warmed M16 medium under paraffin oil at 37 °C in a 5% $CO_2$ atmosphere. At appropriate time points, oocytes were selected for different experiments. In order to inhibit the spontaneous meiotic resume in vitro culture, M2 medium with 50 μM IBMX was used to maintain the GV stage.

### Nocodazole treatment and recovery
To destroy the spindle apparatus, wild-type MI and MII stage oocytes were cultured in M16 medium with 20 μg/ml of nocodazole for 20 min. To re-assemble spindle microtubule, nocodazole was washed out and oocytes were cultured in fresh M16 medium for 30 min.

### Microinjection of OLA1 siRNA
OLA1 siRNA (sc-145833; Santa Cruz, CA) was used to knockdown OLA1. Control siRNA (sc-37007; Santa Cruz, CA) was used as negative control. For microinjection, 5 pL of 30 μM control siRNA and OLA1 siRNA were injected into fully grown GV oocytes, to completely degrade the targeted mRNA, oocytes were cultured in M2 medium with 50 μM IBMX for 24 h. After that, oocytes were directly collected for western blotting or culture in fresh M16 medium for meiotic maturation.

### Immunofluorescence analysis
Oocytes were fixed and permeabilized in PBS containing 4% paraformaldehyde, 0.5% Triton X-100 (pH = 7.4) for 40–50 min. Then, oocytes were blocked in PBS containing 2%

BSA 1 h at room temperature and incubated overnight at 4 °C with anti-OLA1 antibody (ABclonal, 1:100), anti-$\alpha$-tubulin-FITC antibody (Sigma, 1:100). After washing 3 times in PBS containing 0.05% Tween-20, oocytes were incubated with appropriate secondary antibodies for 1 h at 37 °C. Chromosomes were visualized by staining with 1 µg/ml of DAPI for 5–10 min at room temperature. Finally, oocytes were mounted on slides and immunofluorescent images were observed by a confocal laser scanning microscope (Zeiss LSM 800, Germany). Non-immunized rabbit or mouse IgG was used as negative control.

## Western blot

Oocytes were placed in 2X SDS loading buffer and boiled for 5 min at 95 °C. The proteins were separated by SDS-PAGE and then electrophoretically transferred to polyvinylidene fluoride (PVDF) membranes. After transfer, the PVDF membranes were blocked in TBST containing 5% non-fat milk for 1 h, followed by incubation with OLA1 antibody (ABclonal, 1:1,000) overnight at 4 °C. After three times washes in TBST buffer, the membranes were incubated with HRP conjugates secondary antibodies at room temperature for 1 h. After three times washes, bands were visualized using ECL kit. GAPDH antibody was served as a loading control.

## Chromosome spreading and BubR1 staining

Chromosome spreading was done as described previously (*Chen et al., 2018*). Oocytes were treated with Tyrode's buffer (pH 2.5) for about 30 s at 37 °C to remove zona pellucidae. Then oocytes were fixed in drops of speading solution (1% PFA, 0.15% Triton X-100, 3mM DTT in ddH$_2$O, pH = 9.2) on a glass slide. After undisturbed air drying, slides were washed and blocked with PBS containing 1% BSA. Finally, samples were labeled with BubR1 antibody (Abcame, 1:200) and FITC-conjugated donkey anti-sheep secondary antibody (1:200). Chromosomes were stained with DAPI.

## Statistical analysis

Data are presented as mean $\pm$ SEM from at least three independent replicates. Statistical analyses were made with GraphPad. For statistical comparison, Student's t test was used. A value of $P < 0.05$ was considered significant.

# RESULTS

## Cellular localization and expression pattern of OLA1 during mouse oocyte meiosis

Firstly, we investigated the dynamic distribution of OLA1 during mouse oocyte meiosis by immunofluorescent labeling and confocal microscopy. As shown in Figs. 1A–1P, OLA1 was concentrated on the whole cytoplasm at GV stage, after meiotic resume, OLA1 distributed around the chromosomes and co-localized with α-tubulin. At MI and MII stages, OLA1 displayed a spindle-like localization, suggesting that OLA1 may have some connections with spindle assembly. To validate the OLA1 connection with spindle, MI and MII oocytes were treated with nocodazole. After treatment, the spindles were depolymerized and microtubules were evenly dispersed into the cytoplasm in oocytes Figs. 1S–1T. Meanwhile, OLA1 signal also dispersed into the cytoplasm instead of around the

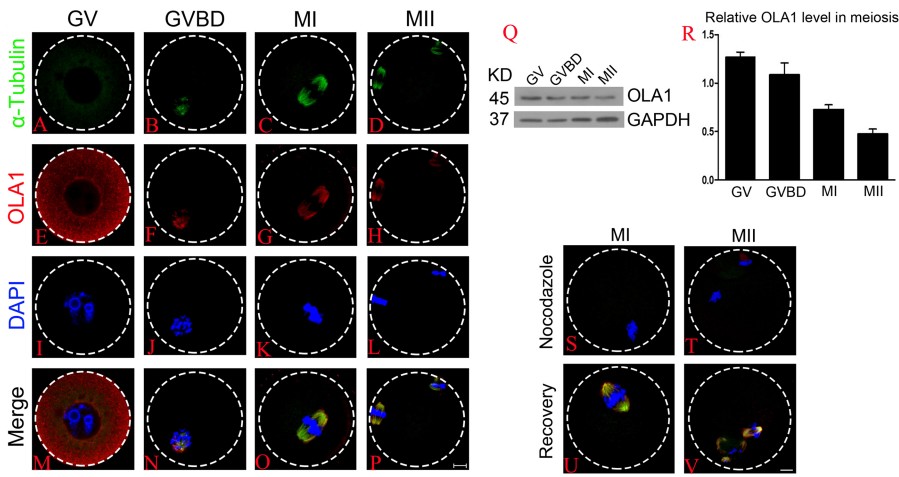

**Figure 1  Cellular localization and expression pattern of OLA1 in mouse oocyte meiosis.** (A–P) Sub-cellular localization pattern of OLA1 in mouse oocyte meiosis was explored by immunofluorescent analysis. Samples were harvested at GV, GVBD, MI, and MII stage and then immunolabeled. α-tubulin, green; OLA1, red and Chromosome, blue. Scale bar, 10 μm. (Q, R) Subcelluar expression pattern of OLA1 in mouse oocyte meiosis was investigated by western blot analysis. Samples at indicated stages were collected. (S–V) Confocal images of OLA1 signal in MI/MII oocytes after treatment with nocodazole and recovery. OLA1, red; α-tubulin, green and chromosome, blue. Scale bar, 10 μm.

chromosomes. Furthermore, as the oocytes were thoroughly washed out of nocodazole and cultured in pre-warmed M16 medium, coupled with spindle re-assembly, OLA1 renewed its spindle-like localization (Figs. 1U–1V), hinting that OLA1 did co-localized with spindle in mouse oocyte. Western blot results showed that OLA1 were expressed at all stages during meiotic progression (Figs. 1Q, 1R).

## Knockdown of OLA1 impairs the germinal vesicle breakdown (GVBD) leading to a decrease in polar body extrusion (PBE)

To further explore the function of OLA1 in oocyte meiosis, OLA1 specific siRNA were injected into GV oocytes. As shown in Figs. 2A and 2B, western blot analysis revealed that expression of OLA1 was significantly decreased when compared with control ($p < 0.05$), suggesting that OLA1- RNAi achieved good knockdown efficiency for the further study of its function in oocytes. GVBD and PBE are two hallmark events in the meiotic progression. We then checked the GVBD and PBE rate after OLA1 knockdown, as shown in Fig. 2C, knockdown of OLA1 significantly inhibited the GVBD progression and the average rate was $86.70 \pm 6.12\%$ in control groups but decreased to $62.83 \pm 1.49\%$ in OLA1- RNAi groups ($p < 0.05$). After 12 h in culture, most of the control oocytes extruded the first polar bodies ($60.38 \pm 6.52\%$), while only $40.58 \pm 2.05\%$ of oocytes extruded the polar body in OLA1-RNAi groups (Fig. 2D, $p < 0.05$). To further confirm whether the decline of PBE was due to the block of GVBD in OLA1- RNAi oocytes, we then analyzed the PBE rate in meiosis-resumed oocytes. As shown in Fig. 2E, once the oocytes underwent GVBD, knockdown of OLA1 had no effect on PBE in meiosisresumed-oocytes ($69.54 \pm 3.45\%$, control vs $64.58 \pm 2.34\%$, $P > 0.05$).

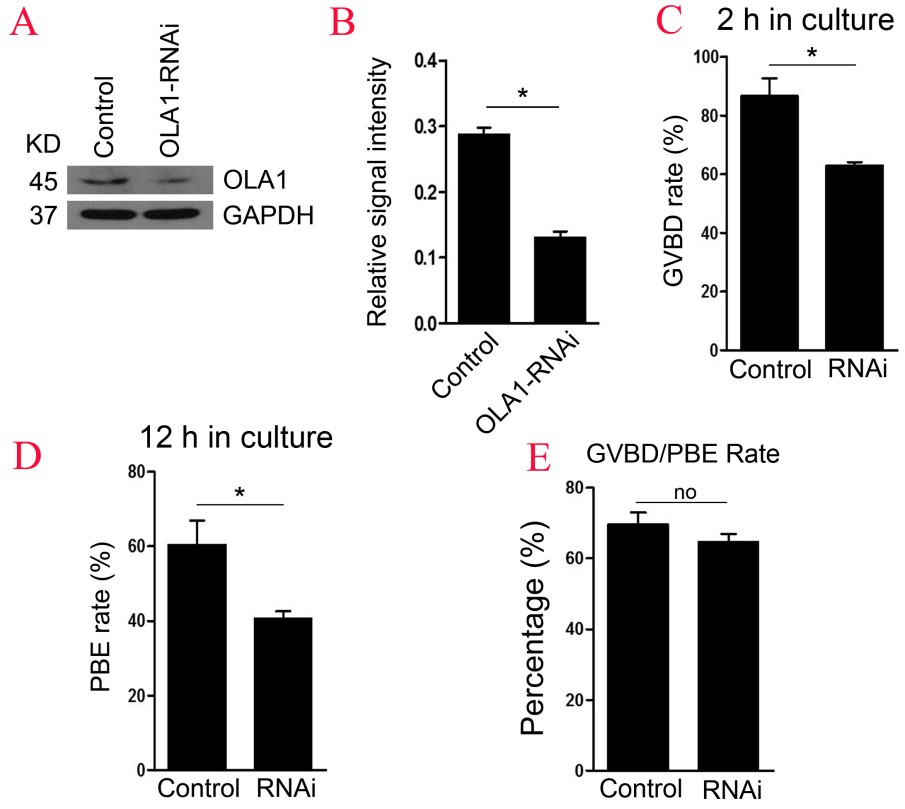

**Figure 2  Knockdown of OLA1 impairs GVBD leading to a decrease in PBE.** Fully grown oocytes injected with OLA1 siRNA or control siRNA were arrested in M2 medium with IBMX for 24 hours, then cultured in vitro for the continuous experiments. (A, B) Knockdown efficiency of OLA1 after OLA1 siRNA was verified by western blot; (C, D) Oocytes after microinjection were released into M16 medium for further culture. The GVBD rate in control group and OLA1 knockdown group were recorded at 2 h; The PBE rate in control group and OLA1 knockdown group were recorded at 12 h; $^*p < 0.05$. (E) PBE rate was characterized in meiosis resumed oocytes in control and RNAi groups. A total of 162 control oocytes and 182 OLA1-RNAi oocytes were calculated.

## Knockdown of OLA1 induces abnormal spindle assembly and chromosome alignment in mouse oocyte

BRCA1 knockdown causes abnormal spindle assembly in mouse oocytes, as a direct interacting protein, OLA1 may modulate the spindle assembly. To verify this speculation, we then checked spindle morphology at metaphase stage by immunofluorescence after OLA1 knockdown. As shown in Figs. 3A–3L, confocal microscopy and quantitative analysis revealed that most control oocytes presented a normal barrel-shape spindle at metaphase stage, while OLA1-RNAi led to abnormal multipolar and small spindles. The proportion of abnormal spindles in OLA1-RNAi group was significantly higher than the control group (30.06 ±0.68% vs. 11.34 ±1.46%, $p < 0.05$; Fig. 3M). Meanwhile, chromosome alignment was also disturbed in OLA1-RNAi oocyte. As shown in Fig. 3I, chromosomes were well-aligned at the metaphase I plate in control oocytes, while OLA1-RNAi oocytes displayed irregularly scattered chromosomes (red arrow, Fig. 3F). The proportion of

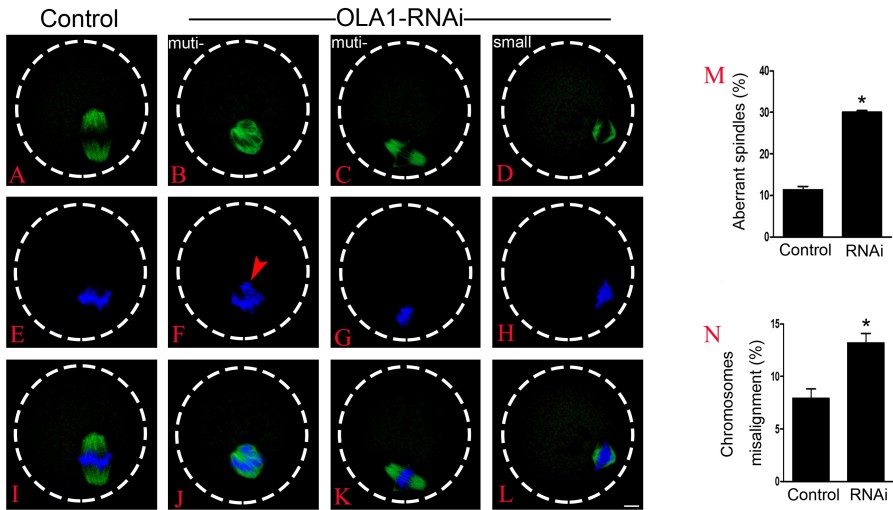

**Figure 3 OLA1 is required for spindle assembly and chromosome alignment in mouse oocytes.** (A–L) Spindle morphology in control and OLA1-depletion oocytes after 6 h post-GVBD culture. α-tubulin, green; chromosome, blue. Scale bar, 10 μm; (M) Quantification of aberrant spindles rate in control and OLA1-RNAi oocytes; (N) quantification of misaligned chromosomes rate in control and OLA1-RNAi oocytes. 88 control oocytes and 90 OLA1-RNAi oocytes were calculated in abnormal spindles and mis-aligned chromosomes. $*p < 0.05$.

misaligned chromosomes in OLA1-RNAi group was also significantly higher than the control group ($13.19 \pm 1.59\%$ vs. $7.92 \pm 1.57\%$, $p < 0.05$; Fig. 3N).

## OLA1 knockdown accelerates the anaphase onset in mouse oocytes

After 6 h post-GVBD in vitro culture, most of meiosis resumed-oocytes in control group have reached at MI (Fig. 4A), while we found some oocytes were extruding the polar bodies in OLA1-knockdown oocytes (red arrow, Fig. 4B). We speculate that OLA1 knockdown could result in premature anaphase onset. To verify this hypothesis, we then did the cycle analysis by α-tubulin staining and confocal microscope. As shown in Figs. 4C–4E, almost all of the control oocytes reached at MI, but we observed a quite number of oocytes have reached at anaphase I or telephase I in OLA1-knockdown oocytes. We also recorded the proportion of oocytes that have reached at MI or A/TI. Results showed that, compared with control, a lower rate of oocytes reached at MI stage in OLA1-RNAi oocytes ($88.00 \pm 3.09\%$ vs. $74.73 \pm 1.27\%$, $p < 0.05$; Fig. 4F), while a significant higher proportion of oocytes has reached A/TI stage ($16.73 \pm 2.19\%$ vs. $2.62 \pm 2.28\%$, $p < 0.05$; Fig. 4F), suggesting that knockdown of OLA1 led to premature of chromosome segregation and accelerated anaphase onset in mouse oocytes.

## OLA1 knockdown causes premature inactivation of SAC

SAC will not abrogate its activity until all the chromosomes are correctly aligned at metaphase I plate, and its previous inactivation can lead to premature chromosome segregation which could finally cause aneuploidy in oocytes. We have found that knockdown of OLA1 induced premature of anaphase onset, we speculated OLA1

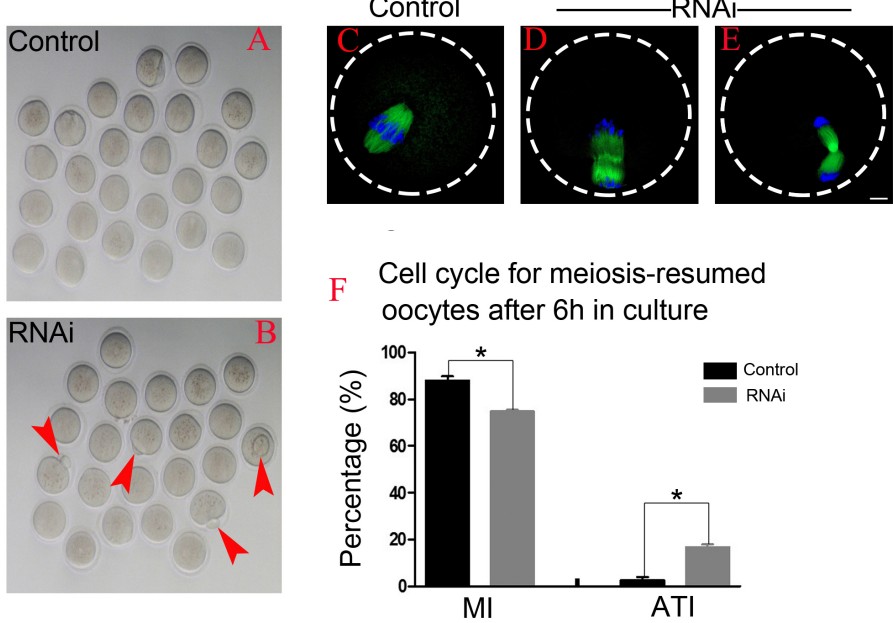

**Figure 4 OLA1 knockdown accelerates the anaphase onset in mouse oocytes.** (A–B) Images exhibited different status of the first polar extraction in control and OLA1-depletion groups; (C–E) the spindle and chromosome morphologies in control and OLA1-knockdown oocytes after 6 h post-GVBD culture. α-tubulin, green; chromosome, blue. Scale bar, 10 μm; (F) cell cycle analysis in control oocytes ($n = 97$) and OLA1-RNAi oocytes ($n = 92$) after 6 h post-GVBD culture. *$p < 0.05$.

knockdown could give rise to the inactivation of SAC, thus we used chromosome spreading and BubR1 (a vital component of SAC signaling) staining to check SAC activity at pre-MI stage (6.5 h) after OLA1-RNAi. As shown in Fig. 5, BubR1 localized to kinetochores at pre-MI stage in all control oocytes, while we could find BubR1 failed to be loaded onto kinetochores from kinetochores in OLA1-depletion oocytes, indicating that depletion of OLA1 caused the previous inactivation of SAC at pre-MI stage.

## OLA1 knockdown promotes the aneuploid rate

As we have observed misaligned chromosomes, premature anaphase trigger and SAC inactivation in OLA1-knockdown oocytes (Figs. 6B, 6C), which all can induce aneuploidy in MII oocyte, so we checked the aneuploid rate at MII oocyte by chromosome spreading, and the result showed that the aneuploid rate was significantly increased in the OLA1-knockdown oocytes ($16.69 \pm 2.51\%$ vs. $26.24 \pm 2.51\%$, $p < 0.05$, Fig. 6D). These results suggest that OLA1 may participate in the SAC activity, thus to contribute to the anaphase trigger and aneuploidy.

## DISCUSSION

Spindle assembly and chromosome segregation are two indispensable events during the progress of meiosis. Defective morphology of spindles and abnormal chromosome alignments could result in aneuploid gametes or the failure of meiosis (*Ma et al., 2014*; *Jiao*

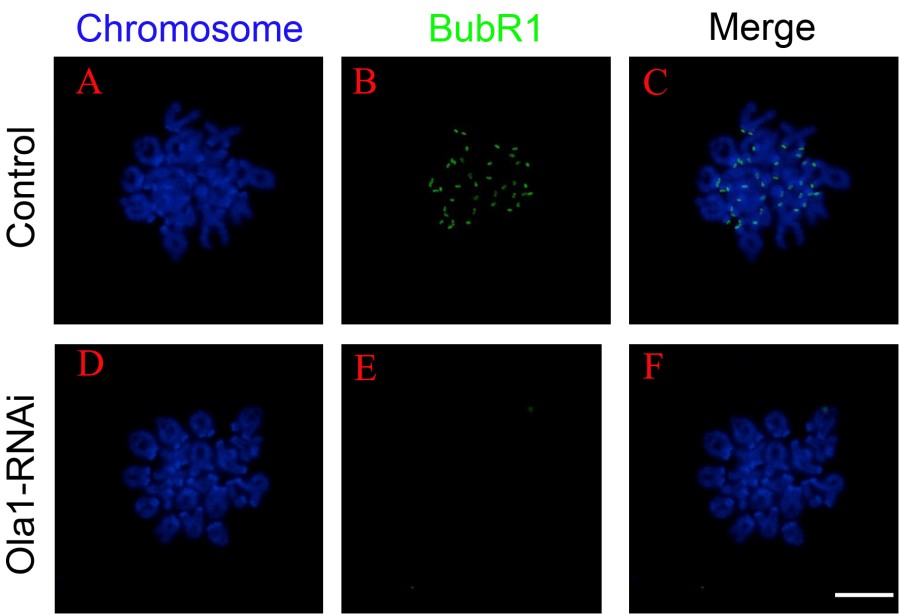

**Figure 5** **OLA1 knockdown causes inactivation of spindle assembly checkpoint.** (A–F) The status of BubR1 at pre-MI stage from control and OLA1-knockdown oocytes. After knockdown, GV oocytes in control and OLA1-knockdown groups were released and cultured in pre-warmed M16 medium for 6.5 h, normally corresponding to Pre-MI stage. Then, oocytes were collected for chromosome spreading and stained with BubR1. BubR1, green; chromosome, blue. Scale bar, 20 μm.

*et al., 2017*; *Han et al., 2015*). There are many factors including cellular protein function to modulate these two processes to be orderly. Currently, the function OLA1 mouse oocyte meiosis is still unknown. In this study, we have discovered OLA1 is a novel participator in oocyte meiosis: the participation in GVBD, spindle assembly and SAC activation.

GVBD and PBE are characteristic features/hallmarks of meiotic progression. When immature oocytes begin to mature by exogenous and endogenous factors stimulation both in vivo and in vitro, GVBD occurs and then oocyte resumes meiosis which could finally bring out the polar body extrusion (*Sánchez & Smitz, 2012*). Defects in GVBD can result in the block of meiosis and oocyte will lose their ability for further development. To detect the function of OLA1, we first examined GVBD and PBE rates after OLA1 knockdown. We found that OLA1 knockdown significantly inhibited GVBD and a decline in PBE was also found in the loss of OLA1. However, once the oocyte underwent GVBD then it could directly extrude the polar body even in the loss of OLA1. And the decline of PBE in OLA1 knockdown oocytes was only due to the block of GVBD. Thus, we conclude that OLA1 is critical for the germinal vesicle breakdown but not polar body extrusion.

Spindle assembly is indispensable for meiotic progression, it supports the segregation of chromosomes in oocytes. Abnormal spindle assembly could result in PBE failure or production of aneuploid gametes (*Schuh & Ellenberg, 2007*; *Clift & Schuh, 2015*; *Bennabi, Terret & Verlhac, 2016*). During the time of spindle assembly, chromosomes show unstable connections with microtubules until hours post GVBD when a tight link between kinetochores and microtubules is established. To avoid the segregation of

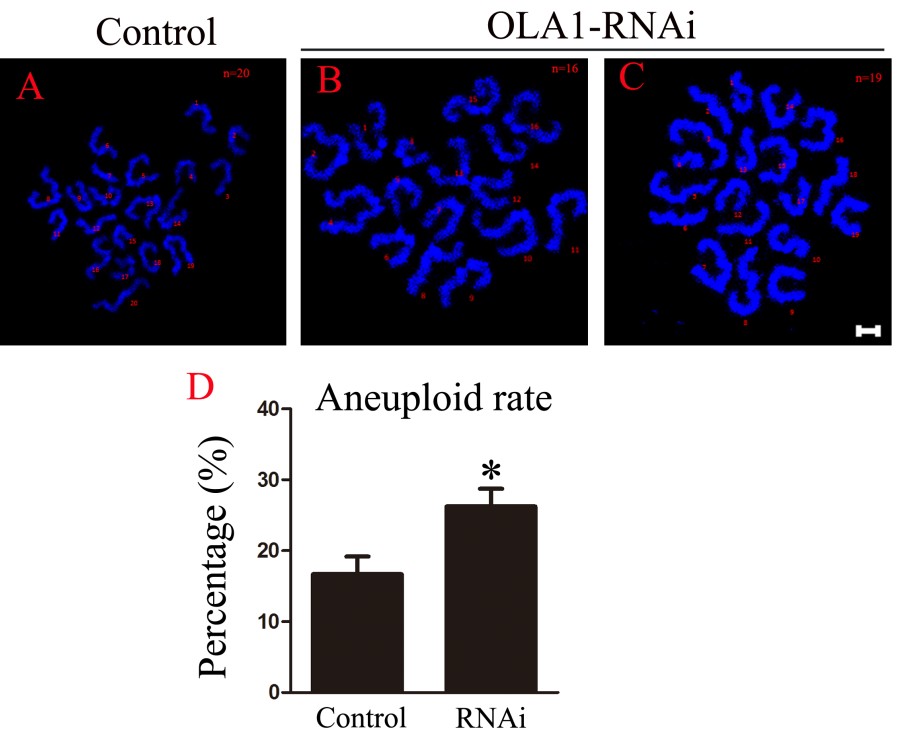

**Figure 6** **OLA1 knockdown promotes the aneuploid rate.** (A–C) Confocal images of chromosome spread of MII oocytes. Control oocytes with a normal haploid complement of 20 chromosomes, OLA1 RNAi oocytes with 16 and 19 chromosomes. Chromosome, blue. Scale bar, 5 μm. (D) The aneuploid rates were recorded in control ($N = 19$) and OLA1 siRNA oocytes ($N = 22$). *$p < 0.05$.

chromosomes with incorrect connection with microtubules, SAC persist at kinetochore (SAC activation) to prevent the premature of chromosome segregation (*Vogt et al., 2008*; *Touati & Wassmann, 2016*; *Sanders & Jones, 2018*). Once all chromosomes align correctly at the metaphase I plate, SAC proteins will dissociated from kinetochores (SAC inactivation) then activates the APC/C activity to accelerate SECURIN degradation and anaphase I onset (*Marston & Wassmann, 2017*). Defects in SAC function can result in incorrect attachments of chromosome with microtubules, and leads to missegregation (*Miao et al., 2017*; *Lu et al., 2017*; *Li et al., 2009*). Study has shown that knockdown of BRCA1 causes abnormal spindles in mouse oocytes (*Xiong et al., 2008*). As an interactive protein in mitosis, we speculated OLA1 may also function in spindle assembly in oocyte. Results showed that multipolar and small spindles occurred in OLA1- knockdown oocytes, indicating that OLA1 also participate in spindle assembly in meiosis. What's more, we speculate that OLA1 could bind to BRCA1 in oocyte exerting its regulatory function in meiotic spindle assembly, but still needs further study to verify that. In addition, a higher proportion of oocytes have reached A/TI stage after 8 h in culture, indicating the premature of anaphase onset in OLA1-knockdown oocytes. Further study identified the inactivation of SAC at pre-MI stage in RNAi oocytes, thus knockdown of OLA1 results in the dysfunction of SAC leading to the premature chromosome segregation. Importantly, knockdown of BRCA1 in mouse

oocyte also leads to the inactivation of SAC (*Xiong et al., 2008*). Taken all these together, it seems that OLA1 may function in meiosis in an OLA1-BRCA1 mediated pathway. In addition, OLA1 knockdown induced spindle defects did not lead to metaphase I arrest, we speculate this might be the bypass of SAC supervision.

## CONCLUSION

In brief, our study highlights that OLA1 exhibits significant function in meiotic progression, especially in GVBD regulation. OLA1 also functions in spindle assembly and SAC activation.

### Funding

This work was supported by the National Key Research and Development Program of China (2016YFC1000600 and 2018YFC1002804), the National Natural Science Foundation of China (81571513, 81771662, 81771618, 81801540), the Major Technological Innovation Projects in Hubei Province (2017ACA101) and the National Key Research and Development Program (2017YFD0501701). The funders had no role in study design, data collection and analysis, decision to publish, or preparation of the manuscript.

### Grant Disclosures

The following grant information was disclosed by the authors:
National Key Research and Development Program of China: 2016YFC1000600, 2018YFC1002804.
National Natural Science Foundation of China: 81571513, 81771662, 81771618, 81801540.
Major Technological Innovation Projects in Hubei Province: 2017ACA101.
National Key Research and Development Program: 2017YFD0501701.

### Competing Interests

The authors declare there are no competing interests.

### Author Contributions

- Di Xie conceived and designed the experiments, performed the experiments, prepared figures and/or tables, approved the final draft.
- Juan Zhang performed the experiments, prepared figures and/or tables, authored or reviewed drafts of the paper, approved the final draft.
- JinLi Ding performed the experiments, prepared figures and/or tables, approved the final draft.
- Jing Yang conceived and designed the experiments, prepared figures and/or tables, authored or reviewed drafts of the paper, approved the final draft.
- Yan Zhang analyzed the data, contributed reagents/materials/analysis tools, authored or reviewed drafts of the paper, approved the final draft.

## Animal Ethics

The following information was supplied relating to ethical approvals (i.e., approving body and any reference numbers):

Animal experiments were approved by the Ethical Committee of the Hubei Research Center of Laboratory Animal (Approval ID: SYXK (Hubei) 2014-0082SCXK). Animals and all experimental procedures were performed according to the guidelines of the Committee of Animal Research Institute (Central Theater General Hospital of PLA, China).

## Data Availability

The raw measurements are available in File S1.

## Supplemental Information

Supplemental information for this article can be found online at http://dx.doi.org/10.7717/peerj.8180#supplemental-information.

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
