# Peer review of "OLA1 is responsible for normal spindle assembly and SAC activation in mouse oocytes"

_PeerJ, doi:10.7717/peerj.8180_

## Round 0.1 · original submission · Major Revisions

Please address the issues raised by all three reviewers for reconsideration.

Reviewer 1 ·

Basic reporting

See below

Experimental design

See below

Validity of the findings

See below

Additional comments

The manuscript entitled ‘OLA1 is responsible for normal spindle assembly and SAC activation in mouse oocytes’ by Xie et al., has investigated the role of OLA1 in meiosis. This is an interesting manuscript that shows a novel role for OLA1 in meiosis.

Overall this manuscript requires significant language editing. There are some concepts that are difficult to interpret and some language is incorrect.

Introduction:

The introduction is brief and could do with some addition information to help guide the reader. For example, little is given about the spindle assembly checkpoint and why this is so important. While it could be a language issue, there is an implication that this is a single event rather than a multi protein effort to ensure correct progression.

The authors mention in the introduction that this GTPase is actually a better ATPase. This is not the case. It is from a family from GTPases (Obg-like) but has evolved to use an alternative nucleotide. OLA1 is mentioned briefly but its importance is not fully defined. It would be beneficial to include additional information about this protein and why the authors decided to study it. Much work has been done to look at the role of OLA1 in centrosome regulation. It has been shown that OLA1 is involved in centrosome focusing and microtuble aster formation (Matsuzawa et al., 2014 Mol Cell 9:101-14), Several sentences are directly taken from the above manuscript, for example: ‘results in the centrosome amplification and the activation of microtubule aster formation’. The authors should re-word the published data and fully explain the importance of this protein. The authors also mention BRCA1 (a known oncogene) but fail to fully address its importance in the context of this study.

Additionally, OLA1 has been shown to be involved in protein synthesis (Chen et al., 2015 Scientific Reps, 5:13241). OLA1 has also been shown to regulate p21 (Ding et al., 2016, Mol and Cell Biol, 36:2568-82). This could be very important as p21 has been shown to inhibit CDK1 (Boufraqech et al., 2106, Oncotarget, 7:29023-35) and be involved in spindle positioning and orientation through p21-activated kinase 4 (Bompard and Morin, 2012, Bioarchitecture, 2:130-33)

Specific points:
Line 68 – GVBD not defined in main body of text.
Line 71 – SAC not defined in main body of text.
Line 81 – NEBD acronym not required and not used elsewhere in the manuscript.

Results:

The results are clearly laid out in a logical order. Line 162 the authors say that localization to spindle means that it is involved in spindle formation – this is not necessarily the case.

Fig 1B (line 170) – The authors state in the main text that OLA1 is consistently expressed throughout meiotic progression, however the blot (representative?) presented shows that there is a 50% reduction over time.

In Fig 1C the authors say that there is a redistribution of OLA1 in cells, however from the images that are presented, this cannot be determined.

Fig 2A/B – The OLA1 siRNA shows that there is a 50% decrease in the amount of protein. This is still a significant amount. Additionally, one assumes that this is a population and therefore some may have a small reduction and others more substantial. This does make interpreting the data difficult. As the protein is still present, perhaps authors should use a reporter based system (i.e. miRNA with bicistronic GFP expression) and cell sort for knockdown cells. This would then allow the authors to be confident of their results and may generate more significant differences.

Fig 2C/D the rates presented are decreased in the RNAi treated cells, however is this a block or just a delay? It is not clear from the data presented.

Authors should explain the importance of the BubR1 (Line 219) experiments.

Can the siRNA phenotype be rescued with reexpression of OLA1?

The authors conclude that OLA1 works in a BRCA1-mediated pathway but there is no data to support this. It would be useful to determine if BRCA1 co-localize. Additionally, is OLA1 localization dependent upon BRCA1?

Discussion/Conclusions:

The data presented here clearly shows that OLA1 is involved in spindle regulation and that in its absence there is faster progression through the cell cycle.

There are several open questions that could be addressed.

One would expect the same results be seen in cells expression E168Q point mutant. While this may not be experimentally determined, it should be addressed in the discussion. Given that this is a breast cancer associated mutation, it would be interesting to see if this would be the same in meiosis.

It should be made clearer that OLA1 may be part of the SAC. Given the additional information mentioned above, it may influence spindle dynamics rather than specifically be part of the SAC.

Overall, this is an interesting study but requires more work to reach some of the conclusions highlighted by the authors. At least, the authors should look at BRCA1 localization and whether OLA1 localization is dependent upon BRCA1.

·

Basic reporting

In this manuscript, Xie et al report the role of OLA1, a member of the GTPase protein family in oocyte meiosis. The authors show that OLA1 can regulate germinal vesicle breakdown (GVBD), spindle assembly and spindle assembly checkpoint (SAC) activation during oocyte meiosis. Transient siRNA knockdown of OLA1 is associated with defects in spindle assembly and SAC inactivation during meiosis. While the authors validate their findings in a clear manner, a few points of concern are mentioned below:
Overall the results support the hypothesis put forward by the authors, but the manuscript needs careful proofreading. The authors report the role of OLA1 in oocyte meiosis. However, there is not sufficient mention or explanation of the oocyte meiotic cycle in the Introduction section. A brief description of the oocyte cycle should be described, explaining the process of GVBD and polar body extrusion (PBE). Sufficient references and background context is provided otherwise.
Earlier studies have already shown that OLA1 is implicated in centrosome amplification and astral microtubule formation during mitosis, associated with spindle defects. The role of OLA1 in meiosis, exhibiting spindle assembly defects would have been predictable to some extent. Thus, in terms of novelty, this manuscript might require additional mechanisms to strengthen their observations.
There are several grammatical errors throughout the manuscript which if corrected will enhance the quality of this article considerably.
Line 59 “Mammalian gametes are yield” is incorrect grammar.
Line 66 “normal progress of oocyte meiosis” should be rephrased as “normal progression of oocyte meiosis”.
Line 71, 216, 263: “premature of chromosome segregation” is again incorrect, should be rephrased as “premature chromosome segregation”.
Line 73 use “established instead of “amended”.
The transition from line 76 to line 77 has no connecting link. The authors might want to atleast write a sentence highlighting why they were interested to study the role of OLA1 in meiosis after describing meiosis in line 76.
Lines 80-84 is directly copied from the reference Matsuzawa et al 2014. The authors need to rephrase the observation in their own language if they are citing a reference.
Line 87, 189, 255: “interactive factor/protein” is not sounding correct. The author means to say OLA1 is an interacting partner, not interactive.
Line 91: “whether OLA1 involve in meiosis” is again grammatically incorrect. It should be “whether OLA is involved in meiosis”
Line 108: “mice firstly intraperitoneally injected” should be “mice were intraperitoneally injected”.
Line 116: “wild type” instead of “wide type”
Line 117: “To reassemble” instead of “To reassembly”
Line 167: “oocytes were thoroughly washed out of nocodazole” is not the correct representation of the sentence. It should be “nocodazole was washed out”.
In the Results section, where the authors describe that silencing OLA1 abrogated GVBD and subsequently PBE, they should describe how did they assess the GVBD and PBE rate?
The transition from line 213 to the subsequent section describing SAC inactivation upon OLA1 knockdown is missing. The authors should begin this section by mentioning why are they looking at SAC activation and its relevance to the context.
Line 228: “processes” instead of “progresses”
Line 232: “cell cycle symbols” should be replaced with some appropriate word. Maybe “GVBD and PBE are characteristic features/hallmarks of meiotic progression”.
Line 244: “During the time spindle assembly” should be “During the time of spindle assembly”.
Line 250-251 needs grammatical error correction and rephrasing.
Line 263: “mouse oocyte” instead of “muse oocyte”
Line 264: “Taken” instead of “Token”.
Line 266-267 needs rephrasing.
The Discussion section of the article is redundant and is simply states the observations reported in the Results section. An appropriate explanation of the phenotypes reported needs to be provided.
The raw data showing the Western blots of OLA1 expression during meiosis needs labelling of what the lanes correspond to.

Experimental design

The authors validate their hypothesis by relevant experiments. They report several significant phenotypes that are characteristic of defective spindle assembly during meiosis. The images shown are representative of the defects mentioned in the article. A few suggestions that the authors might consider are mentioned below:
The authors may want to conduct some colocalization studies with known spindle proteins in mouse oocytes during meiosis. This would further validate that OLA1 colocalizes with the meiotic spindle (Figure 1A).
The blot showing OLA1 expression (Figure 1B) during meiosis should be probed with stage-speficic meiosis markers to validate that particle stage. For example, SCYP3 antibody can be used as a marker for meiosis. Other proteins that are expressed during GV, GVBD, MI and MII should be probed along with OLA1 to validate the corresponding stage of meiosis.
The authors should mention the experimental procedure to measure the GVBD and PBE rate (Figure2).
This article shows several phenotypic defects on spindle assembly upon OLA1 knockdown. However, the authors do not show any rescue experiments by expressing OLA1 to show that the defects can be restored.
The authors show multipolar spindles and chromosome misalignment as a readout of aberrant spindle assembly (Figure 3). They might want to provide some insight into spindle orientation upon OLA1 knockdown by measuring the spindle angle relative to the substratum. Typically, in control cells, the spindle angle should be close to zero, as they are aligned parallel to the substrate. However, during misorientation, there is an increase in the spindle angle.
In the experiment showing anaphase inset in mouse oocytes, was the population synchronized? (Figure 4) It is better to synchronize the cells and then study the meiotic progression to get uniform data set. A flow cytometric analysis showing the progression of a synchronized population of mouse oocytes should generate data that describes anaphase onset better.
Another general question that the authors may want to comment on would be whether the function of OLA1 in maintaining spindle assembly is dependent on the BARD1/BRCA1 complex or it acts independently? It has been shown that OLA1 binds to BARD1/BRCA1 complex to regulate centrosome amplification during mitosis. Mutants of OLA1 that are unable to bind to this complex do not act efficiently to control centrosome number. Thus, it may be worthwhile for the authors to explore the mechanism of action of OLA1 that underlies the defects observed during meiotic spindle assembly.

Validity of the findings

The authors show convincing defects in spindle assembly during meiosis upon silencing OLA1. It is recommended to conduct rescue experiments to check if the observed phenotypic defects can be restored upon OLA1 expression. This will help in complementing he data further.
Also, it would be good if the authors explained the mechanism of action of OLA1. Since it has been reported that OLA1 functions together with BARD1/BRCA1 to control centrosome number, the authors could shed light on whether OLA1 acts cooperatively with this complex to drive spindle assembly. Also, it would be interesting to check if mutants that are not able to bind BARD1/BRCA1 can still exhibit these phenotypic defects.

Additional comments

This data shown in this manuscript is convincing. The authors report interesting data, validating the role of OLA1 in spindle assembly. The experiments performed are relevant and support the hypothesis well. However, some additional experiments, like the rescue of the observed defects in spindle assembly upon OLA1 restoration might be important. Some sections of the manuscript particularly the Discussion seemed redundant and can be rephrased to explain the results better. The Introduction section should provide some background on the meiotic cycle in oocytes. Careful proofreading is required before the final submission as this manuscript has several typos and grammatical errors.

Reviewer 3 ·

Basic reporting

no comment

Experimental design

no comment

Validity of the findings

no comment

Additional comments

The manuscript by Xie et al., investigated the functions of OLA1 in mouse oocyte meiosis. The authors found that OLA1 distributed in the cytoplasm at GV stage and OLA1 co-localized with spindles after GVBD. Knockdown of OLA1 impaired the germinal vesicle breakdown progression. The authors reported that knockdown of OLA1 led to abnormal spindle assembly and loss of OLA1 resulted in the premature segregation of homologous chromosomes. The authors also found that OLA1 knockdown induced premature anaphase onset was due to the precocious inactivation of SAC. The manuscript is well written and provided evidence to support that OLA1 plays important roles in oocyte meiosis. There are some concerns to be addressed.
Comments:
1)The decreased PBE rate in OLA1-knockdowed oocytes resulted from partially inhibited GVBD. Strictly speaking, the phenotype of OLA1 knockdown is impaired GVBD not decreased PBE.
2)The authors could check if aneuploidy exists in OLA1-knockdowed MII oocytes because of premature segregation of homologues.
3)For BubR1 staining, oocytes in stages earlier than pre-MI (6.5h) should be checked to make sure that BubR1 was failed to be loaded or it could be loaded and removed prematurely.
4)The mechanisms of impaired GVBD in OLA1-knockdowed oocytes in further study.

---

## Round 0.2 · accepted · Accept

Revised manuscript is ready for publication.

·

Basic reporting

The authors have addressed all the grammatical errors, pointed out in my previous review comments and the manuscript has been proofread carefully. The manucript reads well after incorporating the amendments suggested in my review comments. However, the authors did not modify the Discussion section to provide a better explanation of the results in the revised draft.

Experimental design

The authors did not include the Western Blot showing the different meiotic stage markers to validate their data. However, they provided an explanation stating that most of the studies use time point to validate the corresponding meiotic stage. The authors have not added experiments that rescue the phenotypic defect caused by OLA1 knockdown and have cited references that have not showed rescue experiments. However, the authors could have tried to conduct a rescue experiment. The authors also have not incorporated any statistical analysis to shed light on the spindle misorientation caused by OLA1 knockdown, as suggested in my review comments. Overall, the authors have not conducted any of the experiments suggested in the review comments. Nevertheless, they have provided references to cite that similar studies have published their results and have provided explanation for not incorporating any additional results.Although the manuscript could have been better if the experiments suggested earlier were done in the revised version but the authors have done a good job in addressing the comments and have provided explanation for the same. They have modified the text which definitely reads better now and is acceptable for publication.

Validity of the findings

See above.

Additional comments

See above

Reviewer 3 ·

Basic reporting

The authors have addressed my concerns.

Experimental design

The authors have addressed my concerns.

Validity of the findings

The authors have addressed my concerns.

Additional comments

The authors have addressed my concerns.